# Experimental methodologies can affect pathogenicity of *Batrachochytrium salamandrivorans* infections

Rajeev Kumar[1], Daniel A. Malagon[1], Edward Davis Carter[1], Debra L. Miller[1,2], Markese L. Bohanon[1], Joseph Patrick W. Cusaac[1], Anna C. Peterson[1], Matthew J. Gray[1]*

**1** Center for Wildlife Health, University of Tennessee Institute of Agriculture, Knoxville, Tennessee, United States of America, **2** Department of Biomedical and Diagnostic Sciences, College of Veterinary Medicine, University of Tennessee, Knoxville, Tennessee, United States of America

* mgray11@utk.edu

**Data Availability Statement:** Data are available in TRACE: https://doi.org/10.7290/QUFV8nuqBd.

## Abstract

Controlled experiments are one approach to understanding the pathogenicity of etiologic agents to susceptible hosts. The recently discovered fungal pathogen, *Batrachochytrium salamandrivorans* (*Bsal*), has resulted in a surge of experimental investigations because of its potential to impact global salamander biodiversity. However, variation in experimental methodologies could thwart knowledge advancement by introducing confounding factors that make comparisons difficult among studies. Thus, our objective was to evaluate if variation in experimental methods changed inferences made on the pathogenicity of *Bsal*. We tested whether passage duration of *Bsal* culture, exposure method of the host to *Bsal* (water bath vs. skin inoculation), *Bsal* culturing method (liquid vs. plated), host husbandry conditions (aquatic vs. terrestrial), and skin swabbing frequency influenced diseased-induced mortality in a susceptible host species, the eastern newt (*Notophthalmus viridescens*). We found that disease-induced mortality was faster for eastern newts when exposed to a low passage isolate, when newts were housed in terrestrial environments, and if exposure to zoospores occurred via water bath. We did not detect differences in disease-induced mortality between culturing methods or swabbing frequencies. Our results illustrate the need to standardize methods among *Bsal* experiments. We provide suggestions for future *Bsal* experiments in the context of hypothesis testing and discuss the ecological implications of our results.

## Introduction

*Batrachochytrium salamandrivorans* (*Bsal*) is an emerging fungal pathogen of global conservation concern [1–3]. *Bsal* has been identified as the causal agent in recent near extirpations of wild fire salamanders (*Salamandra salamandra*) in Belgium and the Netherlands [4, 5], and has been detected in live amphibians in captivity and moving through international trade [6–8]. Controlled experiments where hosts are exposed to *Bsal* under standardized conditions suggest that the pathogen has a broad host range, including several salamander and anuran

**Funding:** This work was partially supported by the National Science Foundation Division of Environmental Biology (EEID Grant # 1814520) and USDA National Institute of Food and Agriculture (Hatch Project 1012932) awarded to MJG and DLM. The sponsors did not play any role in the study design, data collection and analysis, decision to publish, or preparation of the manuscript. URLs: https://www.nsf.gov/funding/pgm_summ.jsp?pims_id=5269 https://nifa.usda.gov.

**Competing interests:** NO authors have competing interests.

families [4, 9]. A central tenet to understanding *Bsal* invasion threat is estimating host susceptibility to infection, and whether host infection progresses to clinical disease hence its pathogenicity [10].

One approach to estimating pathogenicity of etiologic agents is using exposure experiments [11, 12]. Exposure experiments can provide useful ecological insights, such as mechanisms or transmission pathways that drive epidemiology, which can have applied implications. For example, comparing whether pathogenicity of *Bsal* is greater in aquatic or terrestrial environments can provide guidance on where and how to apply disease intervention strategies. Similarly, by exposing salamanders to *Bsal* in water versus direct inoculation could lead to identifying the most important transmission pathways. However, unintended variation in exposure methods, pathogen culturing, testing techniques, and host husbandry conditions could confound inferences made on pathogenicity [13]. For example, Martel et al. [9] inferred that the eastern newt (*Notophthalmus viridescens*) was hyper-sensitive to *Bsal* infection, because 100% of individuals exposed to $5 \times 10^3$ zoospores died within 34 days. Longo et al. [14] reported ca. 50% mortality of eastern newts exposed to the same dose of *Bsal* over 18 weeks, with some individuals clearing the pathogen and surviving to the end of the experiment. Several explanations were offered for the difference in findings between these studies, such as population origin, small sample, and possible co-infection with *B. dendrobatidis* (*Bd*, [14]), which is another pathogenic chytrid fungus responsible for widespread amphibian population declines [15]; however for the most part, differences in experimental design were not discussed. Martel et al. [9] used liquid culture as the inoculum, exposed eastern newts to *Bsal* by pipetting it directly on the dorsum, and housed individuals terrestrially; whereas, Longo et al. [14] used agar plates for culturing *Bsal*, exposed individuals to zoospores in a water bath, and newts could select between aquatic and terrestrial conditions [9, 14]. The number of cell culture passages also differed between studies, which could affect *Bsal* pathogenicity due to genome attenuation or differential gene expression [16, 17]. To detect infection, both studies swabbed individuals; however, Martel et al. [9] swabbed once per week and Longo et al. [14] swabbed once every two weeks. Given that *Bsal* is a skin pathogen and swabbing removes skin cells and chytrid zoospores [5, 18], it is possible that swabbing frequency could affect host-pathogen interactions. Although the North American *Bsal* Task Force provides recommendations for controlled experiments using *Bsal* on their website (http://www.salamanderfungus.org), most of the recommendations are based on expert opinion and investigator preference.

There is a need to evaluate the potential impacts of variation in protocols for *Bsal* susceptibility experiments on host-pathogen interactions. Our objectives were to compare differences in *Bsal* pathogenicity (i.e., likelihood of infection leading to clinical disease and death [19]), between the following treatments: (1) low versus high passage culture, (2) water bath versus pipetted exposure, (3) liquid versus plated culture, (4) aquatic versus terrestrial housing of the host, and (5) skin swabbing frequency (every 6 days, every 12 days, or only at necropsy). We hypothesized that pathogenicity of *Bsal* would be greater with the low passage isolate compared to the high passage isolate due to possible attenuation of the former, pipetted exposure to *Bsal* on the skin would be greater than water bath exposure due to higher likelihood of direct contact with inoculated *Bsal* zoospores, plated cultures would be more than pathogenic than liquid cultures because the former produces more synchronized release of zoospores, aquatic housing would be greater than terrestrial because the latter environment would create more opportunity for zoospore desiccation and reduce within-host reinfection, and increasing swabbing frequency would increase pathogenicity by increasing host stress. Given results from these comparisons, we provide recommendations on designing future *Bsal* experiments. In addition, our experiments for objectives (2) and (4) provided some ecological insights into possible mechanisms of *Bsal* transmission and host resistance.

## Materials and methods

### Methods common among experiments

Below are the methods common for all experiments unless noted otherwise. Sample sizes, *Bsal* doses, host life-stage and swabbing frequencies are in Table 1. Similar to Martel et al. [9] and Longo et al. [14], we used post-metamorphic eastern newts (*Notophthalmus viridescens*) for all experiments. Newts were collected from one site in Knox County, Tennessee, USA (Scientific Collection Permit #1504), and were confirmed to be *Bd* negative prior to the start of each experiment. We performed each experiment at 15˚C in environmental growth chambers, with relative humidity maintained between 80–90%. All water-bath exposed newts were exposed to *Bsal* in 100-mL containers with 1 mL inoculum and 9 mL autoclaved dechlorinated water. The *Bsal* used in our experiments was originally isolated by An Martel and Frank Pasmans from a morbid wild fire salamander in the Netherlands (isolate: AMFP13/1), and had been passaged (i.e., split) in cell culture ca. 20 times (P20) at the start of our experiments. The 200-passage isolate (P200) was maintained in culture and split ca. biweekly, while P20 was cryopreserved and revived for each experiment. We grew *Bsal* on TGhL plates and harvested zoospores by flooding each plate with 7 mL autoclaved dechlorinated water and filtering the suspended zoospores through a 20-um filter to remove zoosporangia. The target exposure dose was prepared by diluting the *Bsal* zoospores in autoclaved dechlorinated water (Table 1). For the exposure route, housing and swabbing experiments, we exposed animals to a single dose of $5x10^6$ zoospores. For passage and culture experiments, we used a lower dose of $1x10^6$ and $5x10^5$ zoospores because we were unable to harvest greater quantities for these experiments. These exposure doses were sufficient to cause infection and induce *Bsal* chytridiomycosis [9]. The control newts used for each experiment were exposed to autoclaved dechlorinated water under identical conditions. After a 24-hr exposure period, we removed the animals from the exposure containers and placed them in housing containers. We housed newts terrestrially in 710-mL plastic containers with a moist paper towel and PVC cover object. The exception was the housing experiment, where half of the newts were housed in circular 2-L containers with 300 mL of dechlorinated water. To minimize accumulation of nitrogenous waste, we transferred newts into clean containers and replaced all the materials every three days. We fed terrestrially housed newts small crickets corresponding to 8% of their body mass when containers were changed. Aquatically housed newts were fed bloodworms. We checked newts twice daily for gross signs of *Bsal* chytridiomycosis (e.g., necrotic lesions, skin sloughing, lethargy), and humanely euthanized individuals that lost righting reflex or at the end of the experiment.

We estimated *Bsal* load at necropsy by swabbing the skin of newts following the standardized protocol used for *Bd* [20], and compared loads among treatments. Genomic DNA (gDNA) was extracted from each swab using Qiagen DNeasy Blood and Tissue kits (Qiagen, Hilden, Germany). We estimated *Bsal* load using *Bsal* singleplex qPCR methods similar to those described in Blooi et al. [21]. All qPCR reactions were amplified using an Applied Biosystems Quantstudio 6 Flex qPCR instrument (Thermo Fisher Scientific, USA). Each swab sample was run in duplicate and considered positive if both replicates amplified within 50 cycles. We also verified that newts were *Bd* negative at the start and end of each experiment using qPCR, because co-infection with *Bd* and *Bsal* can affect host survival [14]. For newts that died during the experiment, we confirmed *Bsal* chytridiomycosis by examining histological cross-sections of hematoxylin and eosin stained epidermal tissue [22]. Representative images of each experimental treatment are provided. We used *Bsal*-induced mortality confirmed by qPCR and histopathology as evidence of pathogenicity [22].

**Table 1. Experiment, treatment, life-stage tested, exposure dose, exposure method, culture type, swabbing frequency, sample size (n), and descriptive statistics (mean and standard deviation, SD) for animals that died or survived the experiment.** Also shown are Wilcox rank-sum and Kruskal-Wallis tests comparing *Bsal* loads of all animals that died during the experiment as well as test results comparing all animals used in each treatment.

| Experiment | Treatment | Life Stage | Exposure Dose | Exposure | Culture Type | Swabbing Frequency | N | Dead Animal Bsal Copies/uL | | Survived Animal Bsal Copies/uL | | Necropsy Copies/uL ~ Treatment for Dead Animals | | Necropsy Copies/uL ~ Treatment for All Animals | |
|---|---|---|---|---|---|---|---|---|---|---|---|---|---|---|---|
| | | | | | | | | μ(N) | SD | μ(N) | SD | W or X^2 | P | W or X^2 | P |
| **Passage** | Control | Adult | Autoclaved Dechlorinated Water | Water Bath | Plate | Every 6 Days | 5 | | | 0(5) | 0 | 4 | 0.08 | 28 | 0.11 |
| | 20X | Adult | 1x10e6 | | | | 10 | 46926.54 (10) | 47720.7 | | | | | | |
| | 200X | Adult | 1x10e6 | | | | 10 | 2078.95(3) | 31515.71 | 16965.51 (7) | 118048.33 | | | | |
| **Exposure** | Control (Water Bath)** | Adult | Autoclaved Dechlorinated Water | Water Bath | Plate | Every 6 Days | 2 | | | 0(2) | 0 | 2 | 0.1 | 14 | 0.6 |
| | Control (Pipette) | Adult | Autoclaved Dechlorinated Water | Pipette | | | 2 | | | 0(2) | 0 | | | | |
| | Water Bath* | Adult | 5x10e6 | Water Bath | | | 6 | 57400.43 (6) | 35511.13 | | | | | | |
| | Pipette | Adult | 5x10e6 | Pipette | | | 6 | 28862.68 (3) | 14059.314 | 59507.97 (3) | 9387.464 | | | | |
| **Culture Type** | Control | Eft | Autoclaved Dechlorinated Water | Water Bath | Water | Every 6 Days | 3 | | | 0(3) | 0 | 3.5 | 0.7 | 16.5 | 0.07 |
| | Liquid | Eft | 1x10e5 | | Liquid Broth | | 8 | 3432.88(2) | 4854.83 | 0(6) | 0 | | | | |
| | Plated | Eft | 1x10e5 | | Plate | | 8 | 9410.3(5) | 12377.91 | 45.99(3) | 79.65 | | | | |
| **Housing** | Control (Aquatic) | Adult | Autoclaved Dechlorinated Water | Water Bath | Plate | Every 6 Days | 2 | | | 0 | 0 | 10 | 0.91 | 15 | 0.7 |
| | Control (Terrestrial)** | Adult | Autoclaved Dechlorinated Water | | | | 2 | | | 0 | 0 | | | | |
| | Aquatic | Adult | 5x10e6 | | | | 6 | 61648.91 (3) | 33928.65 | 36581.91 (3) | 54367.21 | | | | |
| | Terrestrial* | Adult | 5x10e6 | | | | 6 | 57400.43 (6) | 35511.13 | | | | | | |
| **Swabbing Frequency** | Control (6 day swab)** | Adult | Autoclaved Dechlorinated Water | Water Bath | Plate | Every 6 Days | 2 | | | 0(2) | 0 | 3.37 | 0.19 | 1.63 | 0.44 |
| | Control (12 day swab) | Adult | Autoclaved Dechlorinated Water | | | Every 12 Days | 2 | | | 0(2) | 0 | | | | |
| | Control (Necropsy swab) | Adult | Autoclaved Dechlorinated Water | | | Only Necropsy | 2 | | | 0(2) | 0 | | | | |
| | 6 day swab* | Adult | 5x10e6 | | | Every 6 Days | 6 | 57400.43 (6) | 35511.13 | | | | | | |
| | 12 day swab | Adult | 5x10e6 | | | Every 12 Days | 6 | 140061.79 (5) | 99881.34 | 24931.4 (1) | | | | | |
| | Necropsy swab | Adult | 5x10e6 | | | Only Necropsy | 6 | 59445.86 (6) | 37745.56 | | | | | | |

*Indicates the experimental group was used for more than one comparison.

**Indicates the control group was used for more than one comparison.

## Experiment-specific methods

In order to minimize the total number of animals used for these experiments, we used newts exposed via water bath, housed terrestrially and swabbed every 6 days for comparisons of exposure route, housing conditions and swabbing frequency. These newts only differed from the other treatment group by the specific exposure method being compared (Table 1). For example, the water bath and pipette-exposed newts were exposed to the same passage isolate and zoospores harvested from plates, and newts were housed terrestrially after *Bsal* exposure and swabbed every six days.

**Bsal passage history experiment.**   To test whether passage history of cultures affected *Bsal* pathogenicity, we randomly exposed newts to isolates from one of two culture treatments: 20 and 200 passages. We defined a passage as splitting cultures by inoculating 1 mL of TGhL broth containing suspended *Bsal* into 9 mL of new TGhL broth [23].

**Bsal zoospore exposure route experiment.**   We tested if route of zoospore exposure influenced pathogenicity by randomly exposing newts to zoospores in water (as previously described) or by pipette inoculation. The pipette-inoculated newts were exposed by pipetting 1 mL of *Bsal* inoculum onto the dorsal aspect of the newt similar to Martel et al. [9].

**Bsal culture type experiment.**   We tested for differences in *Bsal* pathogenicity between culturing methods by randomly exposing newts to either inoculum collected from TGhL plates (as previously described and done by Longo et al. [14]) or to zoospores harvested from TGhL broth media containing suspended *Bsal* similar to Martel et al. [9]. We filtered the broth media identical to TGhL plates to create the inoculum.

**Housing experiment.**   To test whether housing conditions (terrestrial vs. aquatic) affected pathogenicity, we randomly assigned newts to either terrestrial containers (as described before) or to 2-L containers with 300 mL of aged dechlorinated water and a PVC cover object following the 24-hr exposure to *Bsal*.

**Newt swabbing frequency experiment.**   Lastly, we tested whether swabbing frequency impacted *Bsal* pathogenicity by randomly assigning newts to one of three swabbing frequencies: swabbed only at necropsy, every six days or every 12 days. Swabbing technique was identical among treatments and followed Boyle et al. [20].

## Statistical analyses

We compared median survival rates among treatments for each experiment using Kaplan-Meier analysis and the statistical software R (Version 3.6.1) [24]. We evaluated differences between two or more survival curves at $\alpha = 0.05$ using the "survdiff" function in the survival package [25, 26]. Hazard ratios were calculated using the "coxph" function in the survival package for a robust estimate of the magnitude of treatment differences [25]. We compared copies of *Bsal* DNA per uL extracted from swabs collected at necropsy using Wilcoxon rank sum tests or Kruskal-Wallis tests when comparing multiple groups, because data did not follow a normal distribution. If there were >2 treatments and the Kruskal-Wallis test was significant, we used Wilcoxon tests corrected with the Benjamin and Hochberg adjustment for post-hoc treatment comparisons. All *Bsal* copy comparisons were made using the stats package in R studio [24]. Data and R Code for all analyses are provided in the online supporting information.

## Ethics statement

All husbandry and euthanasia procedures followed recommendations provided by the Association of Zoos and Aquariums and the American Veterinary Medical Association, and were approved by the University of Tennessee Institutional Animal Care and Use Committee

(Protocol #2395). Newts that reached euthanasia endpoints were humanely euthanized via transdermal exposure to benzocaine hydrochloride.

## Results

Survival of eastern newts exposed to the P200 culture was significantly greater than newts exposed to the P20 culture ($X^2$ = 11.4 $P$<0.001; Fig 1A). The odds of an individual dying when exposed to the P20 culture were 7.8X times greater than the P200 culture. Although *Bsal* loads were high in all animals that died ($\bar{x}$ = 36,577; SD = 45,791copies per uL), copies at necropsy did not differ significantly between treatments (W = 28, $P$ = 0.18; S1 Fig).

Survival of newts exposed to zoospores via pipette inoculation was greater than newts inoculated via water bath ($X^2$ = 11.6 $P$<0.001, Fig 1B). The odds of an individual dying when exposed to *Bsal* in a water bath were >100X times greater than pipette exposure on the dorsum. No significant differences in *Bsal* loads at necropsy were detected between these two treatments (W = 14, $P$ = 0.59; S1 Fig).

We detected no differences in survival between liquid cultures and flooded plates ($X^2$ = 1.9 $P$ = 0.13; Fig 1D). *Bsal* loads at necropsy were similar among treatments for animals exposed to zoospores harvested from liquid and plated cultures (W = 16.5, $P$ = 0.07; S1 Fig).

Survival of *Bsal*-exposed newts was significantly greater for individuals housed aquatically compared to those housed terrestrially ($X^2$ = 5.3 $P$ = 0.02, Fig 1C). The odds of an infected newt dying in terrestrial containers were 4X greater than newts housed aquatically. No

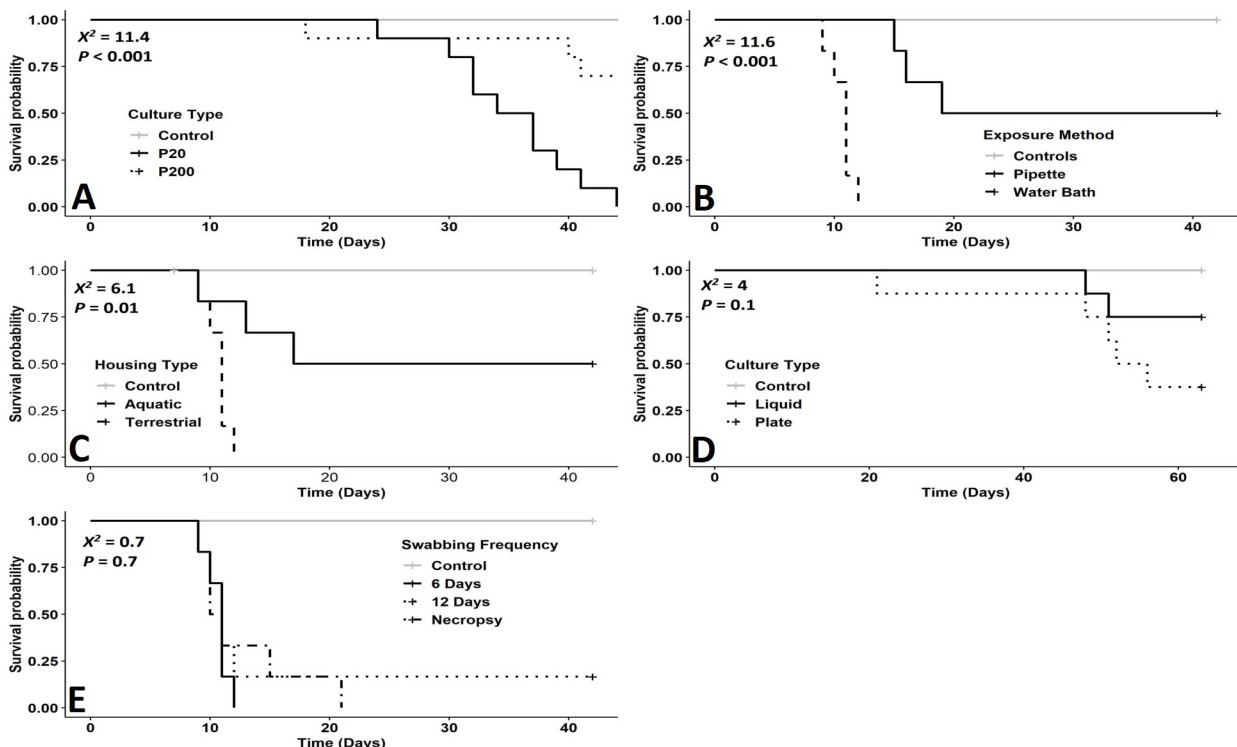

**Fig 1.** (A-E). Kaplan-Meier survival curves showing survival of eastern newts (*Notophthalmus viridescens*) exposed to *Bsal* zoospores. Log-rank test ($\chi^2$) and *P*-values evaluating differences among survival curves for each experiment are shown for animals exposed to P20 or P200 isolates (A), animals exposed to *Bsal* via pipette or water bath inoculation (B), animals housed aquatically or terrestrially after exposure (C), animals exposed to *Bsal* zoospores harvested from TGhL agar plates or TGhL broth (D), and animals swabbed either every 6 days, every 12 days or only at necropsy (E).

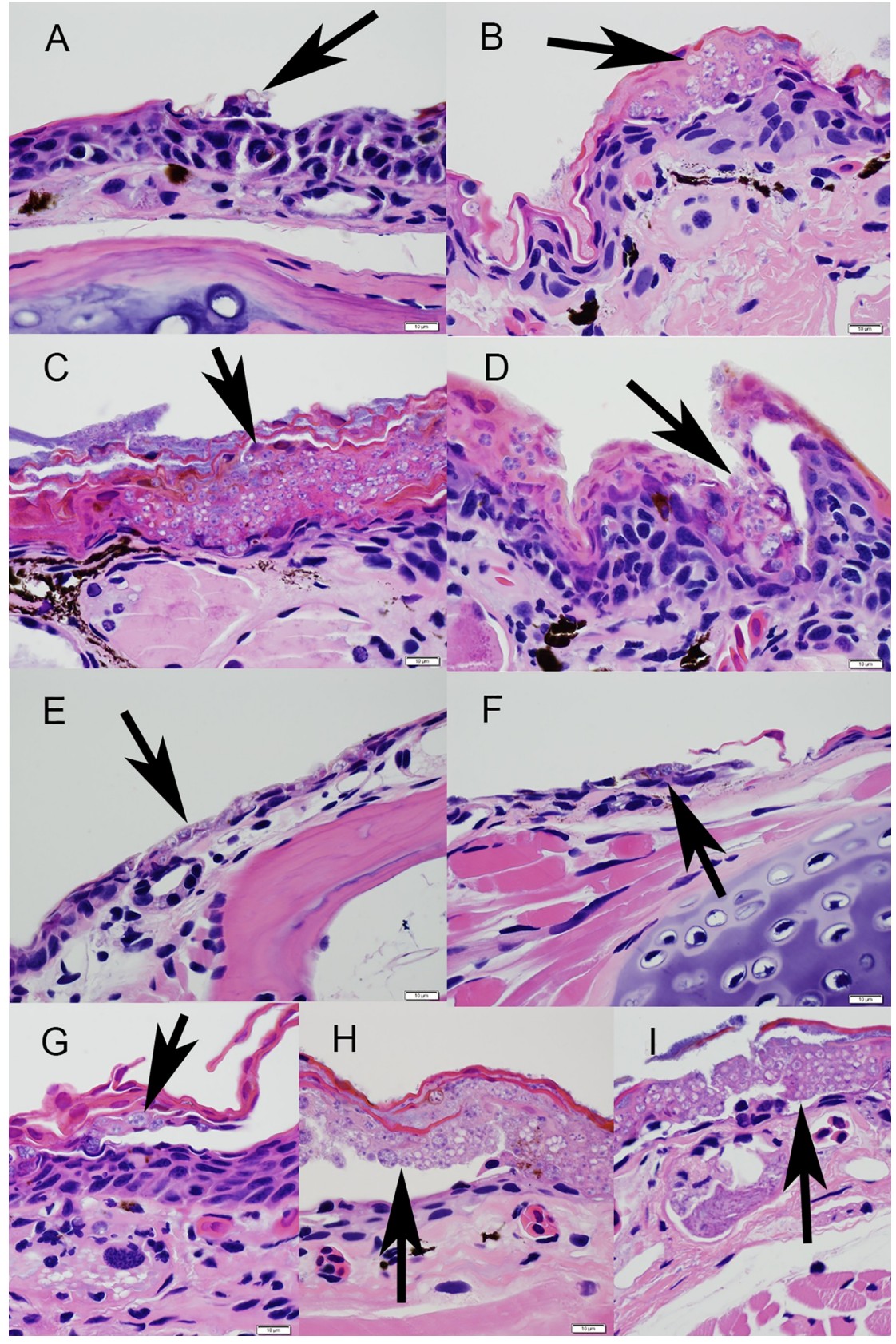

**Fig 2.** Hematoxylin and eosin stained sections of skin from morbid eastern newts (*Notophthalmus viridescens*) showing epidermal invasion by *Batrachochytrium salamandrivorans* (*Bsal*; arrows) for all treatments: passage duration (A = P200, B = P20), water bath (C) vs. pipette exposure (D), plated culture (E) vs. liquid broth (F), aquatic (G) vs. terrestrial husbandry (H), or swabbing frequency (A–G = 6 days, H = 12 days, I = necropsy only). Bar = 10 μm.

differences were detected in *Bsal* loads at necropsy between housing treatments (W = 15, *P* = 0.69; S1 Fig).

We detected no differences in survival among swabbing frequencies ($X^2$ = 0.7 *P* = 0.7; Fig 1E). *Bsal* loads at necropsy were also similar among animals swabbed at different frequencies ($X^2$ = 1.54, *P* = 0.46; S1 Fig).

For all animals that died, we observed histological signs of *Bsal* chytridiomycosis (Fig 2). No control animals died during the study or were qPCR positive for *Bsal* DNA at the end of the experiment. Additionally, no animals tested positive for *Bd* infections at the start or end of the experiment.

## Discussion

We found that *Bsal*-induced mortality was greater for eastern newts when exposed to the low passage isolate, when newts were exposed to zoospores via water bath, and when newts were housed in terrestrial environments. Newts exposed to the P20 isolate had greater odds of dying from *Bsal* chytridiomycosis compared to the P200 isolate, indicating differences in pathogenicity caused by passage history. Several studies on the genetically similar *Bd* chytrid fungus have reported loss of pathogen virulence associated with greater number of passages in culture [16, 17, 27]. Increased passage number can reduce zoospore production rate and total number of zoospores produced by zoosporangia [27]. Although this may have occurred in our study, we controlled for potential differences in production by enumerating zoospores and verifying viability using flow cytometry, and all individuals were exposed to a common dose. Reduced selection (i.e., attenuation) or differential expression of virulence genes in culture could have influenced pathogenicity [17], as suggested by studies comparing *Bd* that was recently isolated from wild hosts to *Bd* in cell culture. For example, Ellison et al. [16]found that *Bd* transcriptomes isolated from two infected amphibian hosts exhibited higher expression of genes associated with increased virulence when compared to a *Bd* culture grown in the lab. Rosenblum et al. [28] also reported that *Bd* cultured on frog skin displayed a greater number of genes coding for proteases that affect pathogenicity when compared to *Bd* cultured using tryptone media [29]. Hence, the differences that we observed in *Bsal*'s pathogenicity may have been driven by genomic changes (e.g., attenuation), phenotypic expression of virulence genes, or shifts in population composition of zoospores to less virulent types in the long-passage isolate.

Newts exposed to *Bsal* via 24-hr water bath had >100X greater odds of dying due to the pathogen than individuals exposed by directly pipetting the pathogen on the animal's dorsum, which may be related to a greater exposed skin surface area in water for pathogen encystment. If so, increased encystment could have led to faster and more severe disease development. It is possible that viability of zoospores pipetted onto the dorsum of an animal also declined more rapidly than zoospores in a water bath, because *Bsal* is predominantly an aquatic pathogen and viability of *Bsal* zoospores decreases rapidly on dry substrate [4]. Thus, infection efficiency of *Bsal* zoospores in water may have contributed to the differences in newts resisting infection. Ecologically, these results suggest that transmission of *Bsal* may be greater in water than through direct transfer of zoospores occurring from host-to-host contact.

Exposure to *Bsal* zoospores collected by flooding TGhL agar plates resulted in greater final mortality (62.5%) than exposure to zoospores grown in and collected from TGhL broth (25%).

Although these differences in mortality were not statistically significant, they represent a 2.5-fold difference in experimental outcomes. Harvesting *Bsal* from TGhL agar plates might more closely resemble the natural life cycle of *Bsal* (i.e., zoospore encysts in the epidermis of the host, forms a zoosporangium, and it releases zoospores [5]). Growing *Bsal* in TGhL broth may represent an alteration from the typical life cycle and select for zoospores and zoosporangia that grow well when immersed in a nutrient solution rather than when adhered to a substrate, including skin. TGhL broth cultures also might result in mixed-aged cultures with fewer infectious motile zoospores compared to more synchronized, even-aged zoospore release on agar plates.

Eastern newts that were housed terrestrially had 4X greater odds of experiencing *Bsal*-induced mortality than those housed aquatically. Although adult eastern newts can be found in terrestrial environments [30, 31], this age class is found most often in aquatic systems [32]. Our study animals were collected from a permanent wetland (i.e., pond), thus the terrestrial environment may have resulted in greater host stress. Increased stress can compromise immune function and thus potentially facilitate greater zoospore infection and disease progression [33]. We also observed that aquatically housed newts were able to shed their skin more easily, which may decrease infection loads and thereby reduce the severity of chytridiomycosis because zoospores are shed into the environment rather than being confined to the animal's skin. Skin shedding has been hypothesized as a resistance mechanism for chytrid infections [34], because shed skin can contain infectious zoospores hence possibly reduce reinfection of the host. Ecologically, the greater pathogenicity of *Bsal* in the terrestrial environment suggests that if infected newts leave the aquatic environment (e.g., during pond drying), their likelihood of dying from *Bsal* chytridiomycosis will increase. Species that mostly or entirely use the terrestrial environment (e.g., *Salamandra salamandra*) also may be at greater risk.

Lastly, swabbing frequency had no apparent effect on survival or *Bsal* loads. Although swabbing can remove zoospores [18], it likely does not remove all zoosporangia, which can extend deeper into the stratum corneum and stratum granulosum [35, 36]. In histological cross-sections, we observed removal of epidermal layers, presumably from swabbing, and the presence of zoosporangia thereafter. Although we did not measure indicators of stress response, it is likely that newts which were never swabbed experienced less stress than swabbed individuals; however, perhaps acute presence of immunosuppressive stress hormones, associated with handling, were offset by some zoospore removal during swabbing [33, 37].

Collectively, our results might provide some additional insight into the differences observed between Martel et al. [9] and Longo et al. [14]. In particular, the isolate used by Martel et al. [9] was lower passage and they housed newts terrestrially; whereas, newts had a choice between aquatic and terrestrial environments in Longo et al. [14]. These methodological differences between the two studies might explain why Martel et al. [9] observed greater mortality than Longo et al. [14] even though the same species was challenged. Interestingly, Longo et al. [14] exposed newts to zoospores in a water bath, yet observed less mortality than Martel et al. [9] who pipetted the pathogen on the dorsum of newts. Hence, exposure method might have less of an effect on *Bsal* pathogenicity than isolate passage duration and housing conditions.

Our results highlight the importance of standardizing methods in *Bsal* experiments if results are going to be compared among studies, or at a minimum acknowledging how methodological differences could lead to biases in interpreting disease outcomes. Given our results, we provide suggestions for future *Bsal* exposure experiments. We recommend that low-passage (<20 passages) inoculum be used for all experiments to facilitate study comparisons, unless the objective is to understand *Bsal* evolution or gene expression in culture. We also recommend flooding TGhL agar plates to collect zoospores, and that the exposure route be chosen to mimic the most likely transmission pathway in nature. For example, transmission of

*Bsal* in fire salamanders likely occurs most often during terrestrial breeding events via contact [4, 38]. Hence, pipetting inoculum on the animal might represent the most realistic route of exposure as it more closely mimics a direct contact scenario. Exposure to *Bsal* in a water bath likely represents a common transmission pathway for aquatic species such as adult eastern newts. Similarly, we recommend that the housing conditions represent the most likely environment of the host, and for hosts that use both environments, the option to enter and leave water should be provided. Lastly, we recommend that the standardized swabbing protocol for *Bd* is followed [18]; however, swabbing frequency should depend on the study objectives. For studies where tracking infection dynamics is essential, swabbing once per week should capture changes in prevalence and loads given that the *Bsal* infected animals typically survive for several weeks [9, 10], allowing for load comparisons over time. However, swabbing can affect histological interpretation of disease progression by removing skin layers (DLM, person. observ.). Given that swabbing frequency did not impact *Bsal*-induced mortality in our study, swabbing a subset of individuals for infection data and using a different set of non-swabbed animals for histological examination and disease determination might be an appropriate methodological design.

One caveat of all methods used throughout this series of challenge experiments is that they do not necessarily reflect the conditions amphibians experience as they encounter pathogens in a natural environment. However, in order to understand the complexities of natural disease systems, it is often useful to evaluate possible factors individually and in combination with a controlled, common-garden experimental design then scale-up influential factors to mesocosm or natural experiments. Reducing methodological differences among controlled studies increases the likelihood that outcomes observed reflect true biological processes.

## Supporting information

**S1 Fig.**
(TIF)

**S2 Fig.**
(JPG)

**S3 Fig.**
(JPG)

**S4 Fig.**
(JPG)

## Acknowledgments

We would like to acknowledge Dr. Jeffrey Kovac and the University of Tennessee College Scholars Program for providing support to DAM. We thank Dr. Bobby Simpson and Alex Anderson of the University of Tennessee East Tennessee Research and Education Center for laboratory and logistical support. We also thank Brian Gleaves, Ciara Sheets, Ana Towe, and Bailee Augustino for assistance with animal care and data collection. Appreciation is extended to Allison Byrne and two anonymous reviewers for providing helpful comments on earlier versions of our manuscript.

## Author Contributions

**Conceptualization:** Rajeev Kumar, Daniel A. Malagon, Edward Davis Carter, Debra L. Miller, Matthew J. Gray.

**Data curation:** Edward Davis Carter.

**Formal analysis:** Edward Davis Carter.

**Funding acquisition:** Matthew J. Gray.

**Investigation:** Rajeev Kumar, Daniel A. Malagon, Edward Davis Carter, Markese L. Bohanon, Matthew J. Gray.

**Methodology:** Rajeev Kumar, Daniel A. Malagon, Edward Davis Carter, Debra L. Miller, Markese L. Bohanon, Joseph Patrick W. Cusaac, Matthew J. Gray.

**Project administration:** Debra L. Miller, Matthew J. Gray.

**Resources:** Debra L. Miller, Matthew J. Gray.

**Supervision:** Debra L. Miller, Joseph Patrick W. Cusaac, Anna C. Peterson, Matthew J. Gray.

**Validation:** Debra L. Miller, Matthew J. Gray.

**Visualization:** Edward Davis Carter.

**Writing – original draft:** Rajeev Kumar, Daniel A. Malagon, Edward Davis Carter, Debra L. Miller, Markese L. Bohanon, Joseph Patrick W. Cusaac, Anna C. Peterson, Matthew J. Gray.

**Writing – review & editing:** Rajeev Kumar, Daniel A. Malagon, Edward Davis Carter, Debra L. Miller, Anna C. Peterson, Matthew J. Gray.

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
