## [Decision Letter · Decision Letter 0]

30 Jul 2020

PONE-D-20-17817

Experimental methodologies can affect pathogenicity of Batrachochytrium salamandrivorans infections

PLOS ONE

Dear Dr. Gray,

Thank you for submitting your manuscript to PLOS ONE. After careful consideration, we feel that it has merit but does not fully meet PLOS ONE’s publication criteria as it currently stands. Therefore, we invite you to submit a revised version of the manuscript that addresses the points raised during the review process.

The reviewers and I agree this revision is a great improvement on the first submission, and acceptable for publication. They also suggest some minor revisions to increase the impact of the paper. I recommend you address these suggestions prior to acceptance. (And, apologies to the authors for the delayed decision, this revision got to me in the middle of a move.)

We look forward to receiving your revised manuscript.

Kind regards,

Wendy C. Turner

Academic Editor

PLOS ONE

Journal Requirements:

Reviewers' comments:

Reviewer's Responses to Questions

**Comments to the Author**

1. Is the manuscript technically sound, and do the data support the conclusions?

Reviewer #1: Yes

Reviewer #2: Yes

2. Has the statistical analysis been performed appropriately and rigorously? 

Reviewer #1: Yes

Reviewer #2: Yes

3. Have the authors made all data underlying the findings in their manuscript fully available?

Reviewer #1: Yes

Reviewer #2: No

4. Is the manuscript presented in an intelligible fashion and written in standard English?

Reviewer #1: Yes

Reviewer #2: Yes

5. Review Comments to the Author

Reviewer #1: I commend the authors for their thorough revisions and believe the paper is much improved. My only remaining concern is that the introduction still lacks context, and as the introduction stands now the reader has no way of knowing a) why these five experimental objectives are relevant or b) what the underlying hypotheses/expectations are. I understand that these details are present in the discussion, but that is read last. An important purpose of an introduction is to establish context for the reader to understand the questions and objectives in the study before diving into the methods and results. However, I will leave it to the editor to decide whether to request this revision.

Reviewer #2: This manuscript is much improved and with a few minor changes I think could be accepted for publication at PLOS ONE. I very much appreciate the addition of Table 1 and the additional details in the methods sections which serve to clarify the experimental design. I also think that changing the language from “tolerance/resistance” to pathogenicity is appropriate and addresses many of the initial reviewer comments. However, I do feel that some of the comments raised by reviewer 1 could be addressed more thoroughly in the text, and not simply in the rebuttal letter.

First, the supplemental figure provided for reviewer 1 could be provided as a supplemental figure to accompany the text and address many of the reviewer comments asking for more information from the Bsal qPCR results. However I am slightly confused by one comment in the rebuttal letter that states the following “Response: We did not process all the swab samples collected as part of this experiment due to limited funds and these data weren’t necessary for inferences on pathogenicity.” This fact is not reported in the paper at all. Rather the methods section implies that all swabs were tested. Which swabs were tested and which were not? How did you test for significant differences in pathogen loads if you did not test all the swabs? This point must be clarified and transparently reported in the text.

Additionally, this paper still fails to connect the findings to a broader significance beyond recommendations for Bsal infection experiments, which I think is a missed opportunity. First, this narrow framing is manifest in the introduction which – as initially stated by reviewer 1 – is underdeveloped. I think the second paragraph of the introduction should be a broader discussion of why some of the factors such as passage number, type of enclosure, and inoculum method would affect pathogenicity and how understanding these effects are relevant to natural systems/conservation. Then the authors can get more specific and discuss the examples they cite.

At line 257 the authors could take an additional step and comment on how the infection efficiency of Bsal zoospores in water is relevant in an ecological or conservation context. Additionally at Line 276 the same comment applies. Discuss the tradeoffs between being able to shed your skin/shed zoospores into the environment vs the risk of being exposed to the pathogen in the water. This study does have broader implications for understating Bsal dynamics in natural settings and I think the authors should try to make some of these connections.

Other specific notes:

The data availability should have provide a unique URL to the specific data.

Line 138 – It is unclear what you mean by “method” here. Can you specify?

Line 191 – I prefer listing the SD along with the mean.

Line 220 – Typo in this sentence “of” should be “or”

6. PLOS authors have the option to publish the peer review history of their article (what does this mean?). If published, this will include your full peer review and any attached files.

Reviewer #1: No

Reviewer #2: **Yes: **Allison Q Byrne

---

## [Author Response · Author response to Decision Letter 0]

31 Jul 2020

All requested changes were addressed. Please see the Response to Reviewers document.

---

## [Editor Report · Decision Letter 1]

25 Aug 2020

Experimental methodologies can affect pathogenicity of Batrachochytrium salamandrivorans infections

PONE-D-20-17817R1

Dear Dr. Gray,

We’re pleased to inform you that your manuscript has been judged scientifically suitable for publication and will be formally accepted for publication once it meets all outstanding technical requirements.

Kind regards,

Wendy C. Turner

Academic Editor

PLOS ONE

---

## [Editor Report · Acceptance letter]

2 Sep 2020

PONE-D-20-17817R1 

Experimental methodologies can affect pathogenicity of *Batrachochytrium salamandrivorans* infections 

Dear Dr. Gray:

I'm pleased to inform you that your manuscript has been deemed suitable for publication in PLOS ONE. Congratulations! Your manuscript is now with our production department. 

Kind regards, 

on behalf of

Dr. Wendy C. Turner 

Academic Editor

PLOS ONE